# Semi-Supervised 3D U-Net Learning Based on Meta Pseudo Labels

Chuda Xiao[1,2,#,], Zhuo Chen[2,#,], Haoyu Li[1,#,], Dan Li[2,], Rashid Khan[1,], Jinyu Tian[2,], Weiguo Xie[2,*,], and Liyilei Su[1,*,]

[1] College of Big data and Internet, Shenzhen Technology University, Shenzhen 518188, China.
[2] Wuerzburg Dynamics Inc., Shenzhen 518118, China.
2070416011@stumail.sztu.edu.cn, zhuo.chen@wuerzburg-dynamics.com,
2100411010@stumail.sztu.edu.cn, dan.li@wuerzburg-dynamics.com,
Rashidkhan@mail.ustc.edu.cn, jinyu.tian@wuerzburg-dynamics.com,
weiguo.xie@wuerzburg-dynamics.com, suliyilei@sztu.edu.cn.

**Abstract.** Deep learning models have demonstrated promising performance for segmenting medical images and are significantly dependent on a huge amount of well-annotated data. However, it is difficult to get a large amount of data, particularly in clinical practices. Likewise, high-performance deep learning models have an enormous model size, restricting their use in actual applications. In order to reduce the burden of both expensive annotations and computational expenses, we designed the semi-supervised knowledge-based method on top of 3D U-Net and Meta Pseudo Labels. We train the teacher network with labelled data to generate the pseudo labels. And then we train the student network on the pseudo labels, and give the training feedback to the teacher network. The student network on FLARE2022 grand challenge Dataset achieved 81.19 % of DSC and 85.20% of NSD. As for the network inference speed, it needs 50.59s for a single case.

**Keywords:** Meta Pseudo Labels · Semi-Supervise · One-step gradient

## 1 Introduction

Abdomen organ segmentation is essential and plays an important role in artificial intelligence-based clinical diagnosis and treatment such as organ quantification and surgical planning etc [1]. In order to accelerate such research and developments, Fast and Low-resource semi-supervised Abdominal oRgan sEgmentation in CT (FLARE 2022) challenge has been introduced, that uses semi-supervised settings and focuses on the usage of unlabeled data. Though it is not easy to segment multiple organs automatically. For instance, multiple organs may vary

---

[1] #These authors contributed equally to this work.
[2] Corresponding author: Weiguo Xie(weiguo.xie@wuerzburg-dynamics.com) and Liyilei Su(suliyilei@sztu.edu.cn).

in shape and size, and thus, organ lesions lead to abnormal segmentation. Similalrly, multi-center data with different scan ranges, and high computing resource are required. However, it is also not easy to collect large amount of annotated data. Thus, semi-supervised learning techniques are efficient to generate labeled data from unlabeled data. Therefore, we propose a 3D U-Net Meta pseduo labels (MPL) method [2], which is a semi-supervise learning approach for segmenting multiple abdominal organs. The proposed method has the following features.

1. Using the MPL semi-supervise method with 3D U-Net[3] can perform automatic segmentation of multiple organs in abnormal CTs.
2. Combine the nnU-Net[4] framework, a surpervise learning framework, with the semi-supervise learning.

## 2   Method

We proposed a pair of networks entails a teacher and student network based on Meta Pseudo Labels. In the proposed network, teacher module generates pseudo labels from unlabeled data. Further, the pseudo labels and labeled images are utillized for training the student module. The teacher network receives feedback from the student network regarding their performance for the improvement of their predictions. The schematic diagram of teacher-student network is depicted in Fig. 1.

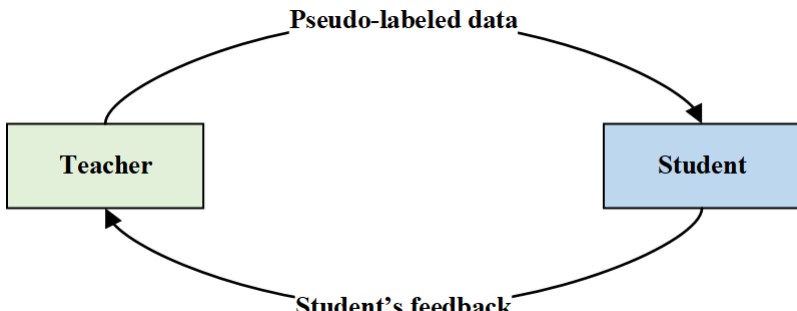

**Fig. 1.** The teacher-student semi-supervised network. When training the models, teacher model would inference the unlabeled data and use the pseudo-labeled data from teacher model to train the student model. The student model provides the feedback to teacher model.

### 2.1   Pre-processing

Our preprocessing approach is the same as the nn U-Net. In the preprocessing step, the computed tomography (CT) images were left uncropped. However,

for anisotropic images (maximum axis spacing $> 3$), in-plane resampling is performed with third order spline whereas out of plane interpolation is performed with nearest neighbor interpolation. Furthermore, we use 0.5 and 99.5 percentiles of the foreground voxels for clipping as well as the global foreground mean (a standard deviation) for normalization on all images.

## 2.2  Proposed method

**One-step gradient.** For the proposed method we formulate our model as follows. Where T and S refer to teacher and student networks, respectively, whereas $w_T$ and $w_s$ refer to their parameters. Let $(x_l, y_l)$ is a batch of images and their labels, and define $x_u$ as a batch of unlabeled images. $p$ denotes the soft prediction and $L$ for the loss function. In our method, the loss function is cross-entropy loss and dice loss.

**The Backbone Network.** Our backbone network utilizes 3D U-Net, which has four upsampling and downsampling layers. Each layer is composed of 3D convolutions, ReLU activations and batch normalization. The first level of the 3D U-Net extracts 32 feature maps and each downsampling process maximizes the extracted feature maps up to 512. The 3D U-Net backbone network is depecited in Fig. 2.

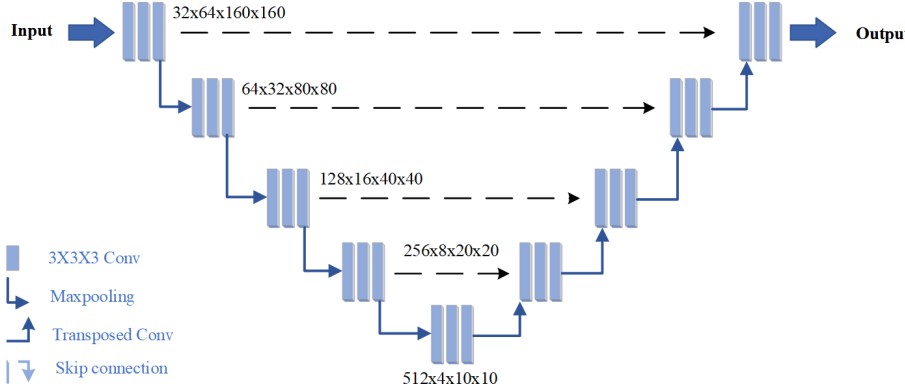

**Fig. 2.** The 3D U-Net network architecture.

**Proposed semi-supervised method.** Our semi-supervised learning method is based on MPL and 3D U-Net as shown in Fig. 3 and their symbolic representation described in One-step gradient(2.2), which having the detail of the method. In the proposed method, the teacher's model $(M_t)$ is trained only by the labeled data which generates the pseudo data. After that, student's model $(M_s)$ uses

---

**Algorithm 1** One-step gradient on MPL

---

1: Input: Labeled data $x_l$,$y_l$ and unlabeled data $x_u$
2: Initialization:$\omega_t^{(0)}, \omega_s^{(0)}, t \leftarrow 0$
3: **repeat**
4:     Sample a labeled example $x_l, y_l$ and an unlabeled example $x_u$
5:     Put $x_l, x_u$ into teacher network get soft prediction $p_l^T, p_u^T$:

$$p_l^T = T(x_l; \omega_T^{(t)}), p_u^T = T(x_u; \omega_T^{(t)})$$

6:     Generate pseudo label $\hat{y}_u$:$\hat{y}_u = Argmax(Softmax(p_u^T))$
7:     Feed $x_l, x_u$ into student network get soft prediction $p_l^S, p_u^S$:

$$p_l^S = S(x_l; \omega_S^{(t)}), p_u^S = S(x_u; \omega_S^{(t)})$$

8:     Update the student using the pseudo label $\hat{y}_u$:

$$\omega_S^{(t+1)} = \omega_S^{(t)} - \alpha_s \nabla_{\omega s} L(\hat{y}_u, p_u^S)$$

9:     Feed $x_l$ into student network get soft prediction: $p_l^{S'} = S(x_l; \omega_S^{(t+1)})$
10:    Compute the teacher's gradient $g_T^{(t)}$ from student's feedback:

$$g_T^{(t)} = \nabla_{\omega_T}(L(\hat{y}_u, p_l^{S'}) - L(\hat{y}_u, p_l^S) \cdot L(p_u^T, \hat{y}_u))$$

11:    Compute the teacher's gradient on labeled data:

$$g_{T,supervised}^{(t)} = \nabla_{\omega_T} L(p_l^T, y_l)$$

12:    Update the teacher: $\omega_T^{(t+1)} = \omega_T^{(t)} - \alpha_T(g_T^{(t)} + g_{(T,supervised)}^{(t)})$
13:    Update epoch: $t \leftarrow t + 1$
14: **until** $t > N - 1$
**Output:** Teacher model $\omega_T^{N-1}$,Student model $\omega_S^{N-1}$

---

the pseudo data and unlabel data to train student model and uses the pseduo labels again by utilizing the gradient descent and update $M_s$ to a new student model i.e., $M_{s+1}$. Finally, we use the output from $M_s$ and $M_{s+1}$ to calculate the student's model feedback which is used as a reward to train the teacher for generating better pseudo labels.

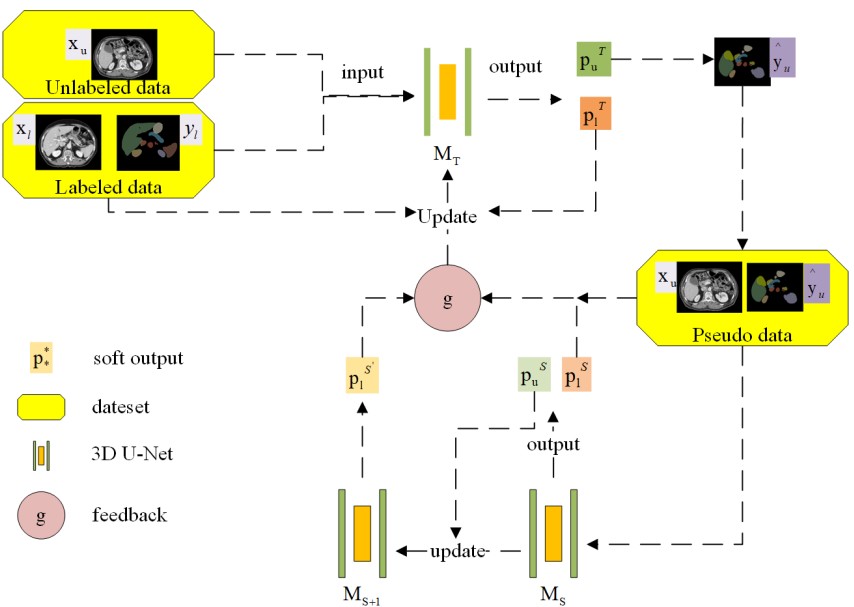

**Fig. 3.** Our proposed semi-supervised method based on the Meta pseudo labels where the backbone model is a 3D U-Net. $x$ denotes the input, $y$ denotes the label or the output, $M$ represents the model, and $p$ denotes the softmax output.

### 2.3  Post-processing

As for post-processing, we just resample the data to its original size.

## 3  Experiments

### 3.1  Dataset and evaluation measures

The FLARE 2022 is an extension of the FLARE 2021 [5] with more segmentation targets and more diverse abdomen CT scans. The dataset is curated from more than 20 medical groups under the license permission including MSD [6], KiTS [7,8], AbdomenCT-1K [9], and TCIA [10]. The training set includes 50 labeled CT scans with pancreas disease and 2000 unlabeled CT scans with liver,

kidney, spleen, or pancreas diseases. The validation set includes 50 CT scans with liver, kidney, spleen, or pancreas diseases. The testing set includes 200 CT scans where 100 cases have liver, kidney, spleen, or pancreas diseases and the other 100 cases have uterine corpus endometrial, urothelial bladder, stomach, sarcomas, or ovarian diseases. All the CT scans only have image information and the center information is not available. The evaluation measures consist of two accuracy measures i.e., Dice Similarity Coefficient (DSC) and Normalized Surface Dice (NSD), and three running efficiency measures: running time, area under GPU memory-time curve (lower than 2048 MB is preferred), and area under CPU utilization-time curve. All measures are used to compute the ranking score.

### 3.2   Implementation details

Our implementation details with respect to configured environments and requirements are provided in Table1. Training protocls for our model are provided in Table2.

**Table 1.** Development environments and requirements.

| | |
|---|---|
| Windows/Ubuntu version | Ubuntu 20.04.1 LTS |
| CPU | AMD EPYC 7742 64-Core Processor |
| RAM | 1.8TB |
| GPU (number and type) | NVIDIA A100 40G($\times$8) |
| CUDA version | 11.4 |
| Programming language | Python 3.8 |
| Deep learning framework | Pytorch (1.10) |

**Table 2.** Training and Inference protocols.

| | |
|---|---|
| Data augmentation | scaling, rotation, random crop, mirror |
| Network initialization | "He" normal initialization |
| Batch size | 8 |
| Patch size | $32\times64\times160\times160$ |
| Total epochs | 1000 |
| Optimizer | SGD with nesterov momentum($\mu$=0.99) |
| Weight decay | 3e-5 |
| Initial learning rate (lr) | 0.01 |
| Lr scheduler | ReduceLROnPlateau |
| Training time | 113 hours |
| Loss function | Dice Loss + Cross Entropy Loss |

# 4    Results and discussions

## 4.1    Quantitative results on validation set

For ablation study to analyze the effect of unlabeled data and semi-supervised learning, the validation set we used was the official data of 50 abdominal CT cases. For labeled data, 50 cases of data were used to train a 3D U-Net model as the baseline segmentation model. For unlabeled data, all of the data were used to train the semi-supervised data model, and the MPL student's model was used to segment organs. The ablation study results are provided in the Table 3, containing the Dice of MPL and baseline method for 13 organs and mean Dice for all classes. Baseline is the 3D U-Net method using all labeled data belonging to supervised learning and MPL is the 3D U-Net method based on proposed MPL, belonging to semi-supervised-based learning. The results show that MPL is better than baseline method.

**Table 3.** Quantitative results on 50 cases of validation set.

| Method | Baseline(Dice) | Baseline+MPL(Dice) |
|---|---|---|
| Liver | 0.9651 | 0.9752 |
| RK | 0.8643 | 0.8840 |
| Spleen | 0.8580 | 0.9390 |
| Pancreas | 0.8530 | 0.8777 |
| Aorta | 0.9510 | 0.9624 |
| IVC | 0.8777 | 0.8909 |
| RAG | 0.8083 | 0.8076 |
| LAG | 0.8100 | 0.8066 |
| Gallbladder | 0.7143 | 0.7540 |
| Esophagus | 0.8794 | 0.8781 |
| Stomach | 0.8633 | 0.8824 |
| Duodenum | 0.7220 | 0.7520 |
| LK | 0.8712 | 0.8499 |
| mean | 0.8491 | 0.8661 |

## 4.2    Qualitative results

Fig. 4 shows successful segmentation of 4 cases including case0002 and case0006 which are easy cases and the two other cases are challenging.The results show that the proposed method can not segment similar organs and organ boundaries well. For example, the left kidney of case0033 was not well segmented, and some of the organ boundaries in case0038 are incomplete.

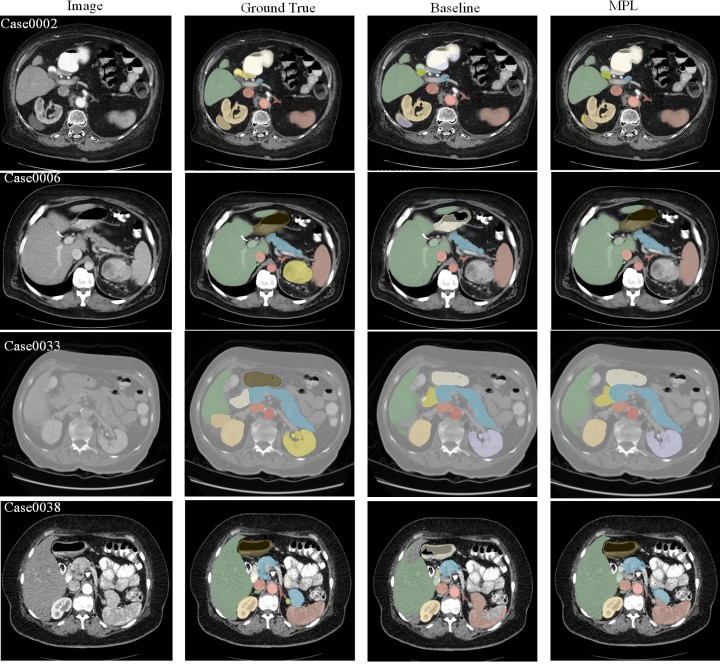

**Fig. 4.** Qualitative results on challenging and easy cases. The case2 and case6 are the easy inference cases , the case33 and case38 are the challenging cases.

### 4.3   Segmentation efficiency results

According to official information, If the GPU memory is less than 2GB when segmenting the organs, the participants would get the prefect score on the AUC_GPU_Time metric, but our model actually used GPU memory which is nearly with 7GB memory. So it indicated that the nnU-Net framework has the defect of too high GPU memory and leads to a relatively large AUC_GPU_Time. The main reason is that model trained by nnU-Net framework is too large, which leads to a long running time and high GPU occupancy.And we did not have any optimization at segmentation efficiency.

### 4.4   Results on final test set

As shown in Table4, we validate our model on the test set from Flare22 and we achieve the Dice score of 0.8119 and the NSD score of 0.8520. As for the efficiency, it costs an average of 50.59s for each case.The AUC_GPU_Time is 178242 and the AUC_CPU_Time is 1221 where infered a case from the official result.

### 4.5   Limitation and future work

**Table 4.** the result of testing on the testing set.

| Method | Dice | NSD |
|---|---|---|
| Liver | 0.93106 | 0.89955 |
| RK | 0.83242 | 0.79604 |
| Spleen | 0.85300 | 0.84863 |
| Pancreas | 0.80521 | 0.89282 |
| Aorta | 0.90872 | 0.92650 |
| IVC | 0.82243 | 0.82277 |
| RAG | 0.82164 | 0.93450 |
| LAG | 0.77253 | 0.88452 |
| Gallbladder | 0.74526 | 0.74285 |
| Esophagus | 0.75769 | 0.84470 |
| Stomach | 0.83977 | 0.85271 |
| Duodenum | 0.68229 | 0.83226 |
| LK | 0.78296 | 0.79772 |
| mean | 0.8119 | 0.8520 |

Our method shows great performance. However, it has limitations, such as high GPU memory usage and long inference time. In the future, we will make efforts to lower GPU usage and speed up the inference.

## 5    Conclusion

In this paper, we proposed a 3D U-Net semi-supervised approach based on Meta Pseudo Labels (MPL) for training the neural networks with limited labeled data and a large number of unlabeled images for medical image segmentation. We trained baseline method using labeled data and 3D U-Net MPL on labeled data and unlabeled data. DSC are used to assess the accuracy. Three different running computational efficiency measures were also computed which proved the effectiveness of our semi-supervised approach experimentally compared to baseline method.

**Acknowledgements** We are thankful to the School-Enterprise Graduate Student Cooperation Fund of Shenzhen Technology University. The authors of this paper declare that the segmentation method they implemented for participation in the FLARE 2022 challenge has not used any pre-trained models nor additional datasets other than those provided by the organizers.

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
