# OpenReview forum: "Semi-Supervised 3D U-Net learning based on Meta Pseudo Labels"
_MICCAI.org/2022/Challenge/FLARE_

### Official Review · Reviewer_TCFC · 2022-09-15

**Rating:** 7
**Confidence:** 5

**Review:**

Nice paper, with good DSC scores.

Some comments/suggestions:

- Why do you only validate on 20 cases? The leaderboard shows your performance on 50 cases.

- Is pre-processing similar or the same as nnu-net, i.e. are you replacing their pre-processing pipeline? Otherwise it is better to write that it is the same.

- You write that you used 1000 unlabelled cases, but there were 2000, why did you not use all?

---

> ### Author Response · Authors · 2022-10-19
> **Thanks for your reviews**
>
> 1. Why do you only validate on 20 cases? The leaderboard shows your performance on 50 cases.
> Response: We are sorry about this typing error, we actually validated on 50 cases and we have modified it in our article.
>
> 2. Is pre-processing similar or the same as nnu-net, i.e. are you replacing their pre-processing pipeline? Otherwise it is better to write that it is the same.
> Response: Yes, the pre-processing is the same as nnu-net, we have added the description in Section 2.1. “Our preprocessing approach is the same as the nn U-Net.”
>
> 3. You write that you used 1000 unlabelled cases, but there were 2000, why did you not use all?
> Response: We are sorry about this typing error, we actually validated on 2000 cases and we have modified it in our article.

---

### Official Review · Reviewer_fNbQ · 2022-09-19
**Good paper**

**Rating:** 8
**Confidence:** 3

**Review:**

- Good scores
- Nice tables, good visualization of segmentation masks.


Lacks a clear description of the method, the motivation behind student feedback needs more explaining. The image describing the method is quite complicated and hard to understand.

The overall method is good, the presentation not so much.

---

> ### Author Response · Authors · 2022-10-19
> **Thanks for your reviews**
>
> 1. Lacks a clear description of the method, the motivation behind student feedback needs more explaining. The image describing the method is quite complicated and hard to understand. The overall method is good, the presentation not so much.
>
> Response: The motivation behind student feedback is used as a reward to train the teacher for generating better pseudo labels. We explained the feedback in Section 2.2 “Finally, we use the output from M_S and M_(S+1) to calculate the student’s model feedback which is used as a reward to train the teacher for generating better pseudo labels.” and modified Fig.3 to make it easier to understand.

---

### Public Comment · ~Jianwei_Gao1 · 2022-09-19
**Good try but the manuscript is not complete enough**

In this paper, the authors designed a semi-supervised knowledge-based method which is on top of 3D U-Net and Meta Pseudo Labels.
Suggestions or deficiencies:
1. It is recommended to use hour instead of day for "Training time" in table 2.
2. It is recommended to use "Dice Loss + Cross Entropy Loss" instead of "Dice + Cross Entropy Loss" for " Loss function" in table 2.
3. The authors should analyze the reasons for the appearance of good cases and bad cases in Figure 4.
4. The authors should add a section of "Limitation and future work".

---

> ### Author Response · Authors · 2022-10-19
> **Thanks for you reviews**
>
> 1. It is recommended to use hour instead of day for "Training time" in table 2.
> Response: We appreciate this comment and we have modified it in Table 2.
>
> 2. It is recommended to use "Dice Loss + Cross Entropy Loss" instead of "Dice + Cross Entropy Loss" for " Loss function" in table 2.
> Response: We appreciate this comment and we have modified it in Table 2.
>
> 3. The authors should analyze the reasons for the appearance of good cases and bad cases in Figure 4.
> Response: We have analyzed the reasons in the Section 4.2. “The results show that the proposed method can not segment similar organs and organ boundaries well. For example, the left kidney of case0033 was not well segmented, and some of the organ  boundaries in case0038 are incomplete.”
>
> 4. The authors should add a section of "Limitation and future work".
> Response: We appreciate this comment and we have added the relevant part in Section 4.5. “Our method shows great performance. However, it has limitations, such as high GPU memory usage and long inference time. In the future, we will make efforts to lower GPU usage and speed up the inference.”

---

### Meta-Review · Program_Chairs · 2022-09-30

**Recommendation:** Major Revision
**Confidence:** 5

**Metareview:**

Reviewer 4:

1. Missing description of post-processing.
2. Missing strategies to improve inference speed and reduce resource consumption. If you do not has any design, please explicitly describe that.
3. Missing limitations and future work.

Reviewer 5:
1. It would be better if the authors could give one or two sentences to describe the "Meta Pseudo Labels" in the abstract.
2. Please provide the results of NSD and efficiency-related metrics in the abstract.
3. Did the authtors leverage any strategy to accelerate the inference procedure?
4. Did the authors exploit any postprocesssing?
5. Please cite "Fast and Low-GPU-memory abdomen CT organ segmentation: The FLARE challenge"
6. Does the Table 4 report the results on the validation set (50 cases)? If not, please provide it.
7. Please discuss the limitation and future work.

Meta review：
Reviewers raise many concerns and suggestions. Please address all comments in the revised manuscript.

---

> ### Author Response · Authors · 2022-10-19
> **Thanks for you reviews**
>
> Reviewer 4:
> 1. Missing description of post-processing.
> Response: We appreciate this comment and we have added the relevant description in Section 2.3. “As for post-processing, we just resample the data to its original size.”
>
> 2. Missing strategies to improve inference speed and reduce resource consumption. If you do not have any design, please explicitly describe that.
> Response: We don’t have any design, and we have mentioned in our manuscript that “And we did not have any optimization at segmentation efficiency” in the Section 4.3.
>
> 3. Missing limitations and future work.
> Response: We appreciate this comment and we have added the relevant part in Section 4.5. “Our method shows great performance. However, it has limitations, such as high GPU memory usage and long inference time. In the future, we will make efforts to lower GPU usage and speed up the inference.”
> Reviewer 5
>
> 1. It would be better if the authors could give one or two sentences to describe the "Meta Pseudo Labels" in the abstract.
> Response: We appreciate this comment and we have added the description in our abstract. “We train the teacher network with labelled data to generate the pseudo labels. And then we train the student network on the pseudo labels and give the training feedback to the teacher network”
>
> 2. Please provide the results of NSD and efficiency-related metrics in the abstract.
> Response: We appreciate this comment and we have added the results in our abstract. “The student network on FLARE2022 grand challenge Dataset achieved 81.19 % of DSC and 85.20% of NSD. As for the network inference speed, it needs 50.59s for a single case.”
>
> 3. Did the authors leverage any strategy to accelerate the inference procedure?
> Response: We have not leveraged any strategy to accelerate the inference procedure.
>
> 4. Did the authors exploit any postprocesssing?
> Response: We have not exploited any strategy to accelerate the inference procedure.
>
> 5. Please cite "Fast and Low-GPU-memory abdomen CT organ segmentation: The FLARE challenge"